# Fractionated Photofrin-Mediated Photodynamic Therapy Significantly Improves Long-Term Survival

**DOI:** 10.3390/cancers15235682

**Published:** 2023-12-01

**Authors:** Hongjing Sun, Weibing Yang, Yihong Ong, Theresa M. Busch, Timothy C. Zhu

**Affiliations:** 1Department of Radiation Oncology, University of Pennsylvania, Philadelphia, PA 19104, USA; hongjing.sun@pennmedicine.upenn.edu (H.S.); weibing.yang@pennmedicine.upenn.edu (W.Y.); theresa.busch@pennmedicine.upenn.edu (T.M.B.); 2Department of Bioengineering, University of Pennsylvania, Philadelphia, PA 19104, USA

**Keywords:** photodynamic therapy, singlet oxygen, Photofrin, PDT explicit dosimetry, macroscopic model, light fractionation

## Abstract

**Simple Summary:**

We have demonstrated that a fractionated PDT scheme, delivered using a two-hour interval between two light delivery treatments, significantly improves long-term efficacy of Photofrin-mediated PDT while reducing side effects. Reactive Oxygen Species Explicit Dosimetry (ROSED), utilizing direct measurements of in vivo light fluence, in vivo Photofrin concentration, and in vivo tissue oxygenation concentration, were used to determine the concentration of reacted reactive oxygen species, [ROS]_rx_, as a dosimetry quantity for PDT treatment. We found that a threshold Photofrin dose of 0.78 mM is required for the complete cure (90-day survival) of RIF tumors using fractionated Photofrin-mediated PDT treatment. To our knowledge, this is the first report of this parameter.

**Abstract:**

This study investigates the effect of fractionated (two-part) PDT on the long-term local control rate (LCR) using the concentration of reactive oxygen species ([ROS]_rx_) as a dosimetry quantity. Groups with different fractionation schemes are examined, including a 2 h interval between light delivery sessions to cumulative fluences of 135, 180, and 225 J/cm^2^. While the total treatment time remains constant within each group, the division of treatment time between the first and second fractionations are explored to assess the impact on long-term survival at 90 days. In all preclinical studies, Photofrin is intravenously administered to mice at a concentration of 5 mg/kg, with an incubation period between 18 and 24 h before the first light delivery session. Fluence rate is fixed at 75 mW/cm^2^. Treatment ensues via a collimated laser beam, 1 cm in diameter, emitting light at 630 nm. Dosimetric quantities are assessed for all groups along with long-term (90 days) treatment outcomes. This study demonstrated a significant improvement in long-term survival after fractionated treatment schemes compared to single-fraction treatment, with the optimal 90-day survival increasing to 63%, 86%, and 100% vs. 20%, 25%, and 50%, respectively, for the three cumulative fluences. The threshold [ROS]_rx_ for the optimal scheme of fractionated Photofrin-mediated PDT, set at 0.78 mM, is significantly lower than that for the single-fraction PDT, at 1.08 mM.

## 1. Introduction

Photodynamic Therapy (PDT) is a clinically approved and minimally invasive therapeutic modality applied in the treatment of both non-oncological ailments and various forms of cancers, entailing an intricate interplay amongst treatment light of specific wavelengths, a photosensitizing agent, and tissue oxygenation dynamics [1,2,3,4,5]. Such interaction generates reactive oxygen species (ROS), leading to cell death while preserving the surrounding tissue of the tumor [6]. Over the past decades, Photodynamic Therapy (PDT) has evolved into a safe and effective treatment in various fields such as dermatology, oncology, and immunotherapy [7,8]. Numerous in vitro studies have been conducted to substantiate PDT as a significant treatment modality, inducing apoptosis in tumor cells [9,10,11,12]. A comprehensive macroscopic model for Reactive Oxygen Species Explicit Dosimetry (ROSED) has been formulated and validated, facilitating the computation of the concentration of reactive oxygen species ([ROS]_rx_), a predictive factor for determining the outcome of PDT treatment [13,14,15]. Conventionally, PDT has been conducted through a single light exposure. However, both pre-clinical and clinical studies have shown improved outcomes through a two-fraction treatment scheme for ALA [16,17,18,19]. Similarly, PDT with Photofrin as photosensitizer has demonstrated heightened efficacy with a different fractionated treatment scheme. Namely, this investigation employed a bloodflow-modulated treatment scheme, reducing fluence rate with increasing blood flow and vice versa. In examining these different fractionation approaches, an intriguing observation was that enhanced treatment outcomes were observed for fractionation schemes with a dark interval of 1 to 2 h between two successive continuous large light fractions, corroborating findings from earlier fractionation studies for ALA [20,21]. Nevertheless, the precise mechanism behind this improvement remains unknown. To further understand the underlying mechanisms of this treatment paradigm and quantify the influence of fractionated light exposure on treatment efficacy, we employed Reactive Oxygen Species Explicit Dosimetry (ROSED). This analytical framework enables the computation of the cumulative reactive oxygen species, [ROS]_rx_, generated by PDT. To comprehensively monitor the cumulative [ROS]_rx_ levels during PDT, we meticulously measured in situ the light fluence rate (ϕ) at the tumor surface, tissue optical properties, the inferred light fluence rate at a 3 mm tumor depth, the in vivo concentration of the photosensitizing agent ([S_0_]), and the in vivo oxygen concentration within the tumor tissue ([^3^O_2_]) at a 3 mm depth throughout the PDT treatment duration. By utilizing this explicit dosimetry approach, we endeavored to quantitatively assess the cumulative impact of fractionated illumination on the therapeutic outcomes of PDT.

While earlier investigations have underscored the improved outcomes of fractionated Photodynamic Therapy (PDT), the majority of these studies focused on 5-Aminolevulinic Acid (ALA), a prodrug that subsequently produces Protoporphyrin IV in tissue. It tends to be effective for superficial tumors rather than deep-seated tumor nodules, potentially rendering it less potent for addressing tumors situated at greater depths [22,23]. Considering this, our current study employs Photofrin^®^ (Porfimer sodium) as the chosen photosensitizer. Photofrin^®^, a partially purified derivative of hematoporphyrin (HpD), boasts a favorable profile of relative non-toxicity and facile formulation [24,25,26]. It received approval from the US Food and Drug Administration (FDA) in 1995 for the management of esophageal cancer and has since received authorization for treating diverse cancer types [27]. Functioning as a type II photosensitizer, when excited by light, Photofrin irreversibly oxidizes nearby molecules by forming highly reactive singlet oxygen (^1^O_2_) through an energy transfer process from its triplet state to the ground state of oxygen (^3^O_2_). This molecular cascade precipitates cytotoxicity and the induction of apoptosis. Additionally, an alternate avenue through which Photofrin-mediated PDT produces tumor destruction is through the damage of the vascular endothelium, leading to erythrocyte leakage and subsequent ischemic tumor necrosis [28,29]. More research is being conducted with the aim of further enhancing treatment efficiency [30,31,32].

To systematically explore the most optimal treatment regimen, this study was split into two parts. Firstly, we determined the efficacy of fractionated Photofrin-mediated PDT in mice with radiation-induced fibrosarcoma (RIF) tumors. The fractionated therapeutic scheme included two PDT treatment sequences, separated by a 2 h intermission. For comparison, non-fractionated PDT with the same total treatment time was performed. The cumulative light dosage for the two light exposures was consistently maintained at 135 Jcm^−2^. Secondly, the results from these studies were further used to inform the design of a fractionation scheme by increasing the total treatment time to increase PDT efficacy, while evaluating the associated toxicities. Toward this goal, different combinations of light fraction durations and total light dosages were evaluated. Dosimetry was performed as the measured light fluence rate (ϕ), photosensitizer concentration ([S_0_]), and tumor oxygenation concentration ([^3^O_2_]). These measured quantities were used to determine the cumulative concentration of reactive oxygen species ([ROS]_rx_). The long-term treatment effect was assessed through a 90-day survival post-PDT.

## 2. Materials and Methods

### 2.1. Tumor Model

Studies were performed in female C3H mice (Charles River Laboratories, Kingston, NY, USA) of ages ranging from 6 to 8 weeks. Radiation-induced fibrosarcoma (RIF) tumors were induced on the dorsal shoulders of the mice through intradermal administration of 3 × 10^5^ cells per mouse. After injecting RIF tumor cells, the dimensions of the tumors were measured daily, and treatment was scheduled when the tumor diameter reached approximately 3–4 mm. The measurements of tumor width (denoted as ‘a’) and length (denoted as ‘b’) were ascertained using a slide caliper, and tumor volume (labeled as ‘V’) was calculated using the formula V = (π × a^2^ × b)/6 [33]. The treatment site was depilated using Nair (Church & Dwight Co., Inc., Ewing, NJ, USA). At ~24 h prior to light delivery, Photofrin was introduced via tail injection, at a concentration of 5 mg/kg. All experimental procedures were executed as described in protocols approved by the University of Pennsylvania Institutional Animal Care and Use Committee. The upkeep and care of the animals was overseen by the University of Pennsylvania Laboratory Animal Resources, in facilities approved by the American Association for the Accreditation of Laboratory Animal Care.

### 2.2. Photodynamic Therapy Protocol

Following a 24 h interval between drug administration and light exposure, a 630 nm laser (Biolitec AG., A-1030, Vienna, Austria) was employed for superficial irradiation. To ensure uniform illumination across the mice’s skin surface, a microlens fiber was coupled to the laser fiber, creating an evenly distributed irradiation with a diameter of 1 cm, encompassing the tumor area. The laser’s output was set at a constant level, such that the in-air fluence rate at the central point of the 1 cm diameter treatment region consistently measured around 75 mWcm^−2^ across all experimental conditions. Before each treatment session commenced, a PDT dosimetry system was utilized to measure the in-air fluence rate at the tumor surface [34]. This practice accounted for the slight individual variability in source-to-surface distances. The experimental framework encompassed a straightforward two-fold treatment regimen, illustrated in Figure 1. Mice were grouped into three categories based on the light dosage, each characterized by a uniform 2 h dark interval that separated the two light exposures. Within each of these categories, several subgroups were established, each marked by distinct combinations of light fraction durations. The study included a control group, comprising mice with RIF tumors that received neither light irradiation nor Photofrin. In the initial phase of the investigation (part 1), a cumulative light dosage of 135 Jcm^−2^ was selected, drawing upon insights gleaned from prior comprehensive animal studies [33]. The experimental groups featured various light fraction durations, including (1) 400 s and 1400 s, (2) 600 and 1200 s, (3) 800 s and 1000 s, (4) 1000 s and 800 s, (5) 1200 s and 600 s, all interspersed with intermediate 2 h dark intervals. In addition, there was a non-fractionated group (1800 s). To further refine treatment efficacy, higher-light dosages (180 Jcm^−2^ and 225 Jcm^−2^) were explored in part 2. This phase encompassed the following treatment schemes: (1) 800 s and 1600 s, (2) 1000 and 1400 s, (3) 1200 s and 1200 s, (4) 1400 s and 1000 s, (5) 1400 s and 1600 s, alongside the non-fractionated group. Furthermore, two additional groups were introduced, involving extended light fraction durations of (1) 2400 s and (2) 3000 s.

### 2.3. Measurement of the Oxygen Concentration in Tumors

In the context of PDT, the in vivo tissue oxygen partial pressure (pO_2_) was meticulously measured by a phosphorescence-based oxygen probe (OxyLite Pro, Oxford Optronix, Oxford, UK). This oxygen probe was positioned within the tumor tissue, penetrating to a depth of 3 mm from the treatment surface prior to PDT. This technique is explained in detail in a previous investigation [13,35]. To calculate the concentration of ground state oxygen ([^3^O_2_]), the measured pO_2_ values were multiplied by the established ^3^O_2_ solubility in tissue, a value quantified at 1.295 μM/mmHg [13]. Owing to the inherent sensitivity of the probe and potential uncontrollable shifts, some degree of fluctuation in the measured [^3^O_2_] was observed. The initial [^3^O_2_] value, designated as [^3^O_2_]_0_, was documented until the measured values reached a stabilized state. Following this stabilization, the treatment protocol was initiated, with subsequent monitoring of ^3^O_2_ unfolding throughout the PDT process, contributing to the post-treatment analytical assessment.

### 2.4. Validation and Measurement of the Photofrin Concentration in Tumors

Throughout the course of PDT, comprehensive measurements were performed to assess the light fluence rate, photosensitizer concentration, and tissue oxygen concentration ([^3^O_2_]). In vivo Photofrin fluorescence spectra were acquired employing a bespoke multi-fiber contact probe, both before and after PDT, at the tumor surface [13,14]. This specialized probe was interfaced with a 405 nm laser (Power Technology Inc., Little Rock, AR, USA) for the excitation of Photofrin fluorescence, coupled with a multichannel CCD spectrograph (InSpectrum, Princeton Instruments, Trenton, NJ, USA) for the collection of the resultant fluorescence spectra. Given the principles governing light propagation in tissue, the focal point of these measurements predominantly corresponds to approximately one-third of the separation distance between the light source and detector. Consequently, the interrogation of tissue takes place at a depth of roughly 0.67 mm beneath the tissue surface. It is imperative to acknowledge that while these measurements provide valuable insights, they predominantly reflect the superficial layer of the tumor, rather than the entire tumor volume. This limitation necessitates an assumption of a homogenous distribution of Photofrin concentration within the tumor. To validate the in vivo measurements obtained using the contact probe, a supplementary ex vivo validation was undertaken. This validation procedure was conducted by comparing the Photofrin concentration assessed in the entire tumor through ex vivo means, as elaborated upon in the subsequent description.

The attenuation of the Photofrin fluorescence signal due to the light absorption and scattering by tissue was corrected by applying an empirical correction factor (CF):(1)$$CF=C_{1}(\mu_{a}^{b_{1}}{\mu^{\prime }}_{s}^{b_{2}}+C_{2})$$
where constants C_1_ = 0.98 ± 0.01, C_2_ = 0.49 ± 0.05, b_1_ = 1.03 ± 0.09, and b_2_ = −0.07 ± 0.03 were determined through fitting using OriginPro 2023b (OriginLab Corp., Northampton, MA, USA), with tissue-simulating phantoms exhibiting different µ_a_ and µ_s_′. Monte Carlo (MC) calculations were conducted to determine the fitting function and its parameters [34]. To ascertain the values of the parameters in the computational expression for the Correction Factor (CF), a series of liquid phantoms was prepared. These phantoms encompassed a range of optical characteristics (μ_a_ = 0–0.7 cm^−1^, μ_s_′ = 2–10 cm^−1^), and consistently featured a constant Photofrin concentration of 5 mg/kg. To construct these phantoms, Intralipid (Fresenius Kabi, Uppsala, Sweden) was employed as the scatterer, while ink sourced from Parker Quink served as the absorber. The analytical insights illustrated in Figure 2a emphasize that the appropriately corrected Singular Value Decomposition (SVD_cor_) values (obtained via SVD_cor_ = SVD_raw_ × CF) should display consistent magnitudes for phantoms characterized by identical Photofrin concentrations. The SVD magnitudes cannot reflect the actual Photofrin concentration. Thus, to confer absolute quantification to the probe-measured in vivo photosensitizer concentration, a calibration curve was established. This calibration process entailed correlating SVD_cor_ values against an array of Photofrin concentrations spanning from 0 to 13.4 μM. Consequently, the fluorescence measurements acquired in vivo could be subjected to SVD fitting, which would subsequently be corrected through the associated CF derived from Equation (1). Thereafter, the absolute Photofrin concentration could be precisely determined by referencing the SVD_cor_ values against the established calibration curve, as depicted in Figure 2a,b. To facilitate the computation of the correction factor within this study, the average optical properties of mouse tissue (μ_a_ = 0.9 cm^−1^ and μ_s_′ = 8.4 cm^−1^, well-established by the previous study [14]) were incorporated to determine a CF of 1.24 ± 0.13. These average properties exhibited congruence within a 10% margin in comparison to individual tissue optical properties, particularly in terms of the projected light fluence rate versus depth, up to a depth of 3 mm at the emission wavelength of 632 nm. The mean optical properties were also applied to calculate the ratio of fluence rate and in-air fluence rate at a 3 mm depth of the tumor using MC simulation [14]. The fidelity of the contact probe, along with the robustness of the SVD fitting methodology, has been previously validated in an assortment of preclinical and clinical investigations, spanning across not only Photofrin but also other photosensitizers [36].

To assess the in vivo photosensitizer concentration, the methods in fluorescence measurements described above were used for all tumors prior to and at the end of PDT. To ascertain the precision and reliability of these in vivo fluorescence measurements, a separate set of mice was used for ex vivo analyses. This parallel investigation involved the determination of Photofrin concentration and its subsequent comparison with the values derived from in vivo measurements. For this study, six mice were administered varying Photofrin doses ranging from 0 to 7 mg/kg. In vivo fluorescence measurements were performed at the surface of each tumor, within a time frame of 18 to 24 h post Photofrin administration. Post these fluorescence measurements, the mice were humanely sacrificed, and their tumors were cautiously excised, safeguarded from light exposure, and stored at a frigid temperature of −80 °C. For the ex vivo assessments, every tumor underwent a process of mincing and division into three distinct samples, thereby facilitating three measurements for each individual tumor. Homogenized solutions of the tumor samples were prepared employing Solvable (PerkinElmer, Waltham, MA, USA). The subsequent fluorescence measurement was executed via a spectrofluorometer (FluoroMax-3; Horiba Jobin Yvon, Inc., Edison, NJ, USA), with an excitation wavelength of 405 nm, coupled with an emission spectrum ranging from 630 to 750 nm, encompassing two emission maxima at 635 nm and 705 nm. The quantification of photosensitizer concentration within the tissue was derived through a calculation based on the shift in fluorescence resultant from the introduction of a known quantity of Photofrin to each sample, after its initial reading. To ascertain the alignment between the ex vivo and in vivo measurements, a linear fitting procedure was employed, leading to the correlation depicted in Figure 2c. The observed agreement between the ex vivo and in vivo Photofrin concentrations, as depicted by the solid line in Figure 2c, underscores the close concordance between these two assessment methods. Each data point within this graph signifies the average outcome of multiple measurements for each individual tumor, while the horizontal and vertical error bars capture the standard deviation characterizing the ex vivo and in vivo measurements.

### 2.5. Macroscopic Singlet Oxygen Model for PDT

In this study, a practical and empirically derived macroscopic Reactive Oxygen Species Explicit Dosimetry (ROSED) model was employed, which draws its foundation from reaction rate equations aligning with the principles of a type II Photodynamic Therapy (PDT) mechanism [37,38]. This macroscopic ROSED model, a powerful tool for gauging the reactive oxygen species concentration ([ROS]_rx_), capitalizes on measured parameters such as light fluence (and its rate), Photofrin drug concentration, and tissue oxygen concentration ([^3^O_2_]). The demonstrated efficacy of this model resides in its ability to streamline the intricate energy transfer processes inherent in type II PDT, ultimately facilitating the calculation of the cumulative [ROS]_rx_ generated during illumination [13,14,36]. In this model, a set of PDT kinetic equations is involved,
(2)$$\frac{d\left[S_{0}\right]}{dt}+\xi \sigma \frac{\phi \left(\left[S_{0}\right]+\delta \right){[}^{3}O_{2}]}{{[}^{3}O_{2}]+\beta }\left[S_{0}\right]=0,$$
(3)$$\frac{d{[}^{3}O_{2}]}{dt}+\xi \frac{\phi \left[S_{0}\right]}{[{}^{3}O_{2}]+\beta }[{}^{3}O_{2}]-g\left(1-\frac{{[}^{3}O_{2}]}{{[}^{3}O_{2}](t=0)}\right)=0,$$
(4)$$\frac{d{\left[ROS\right]}_{rx}}{dt}+\xi \frac{\phi \left[S_{0}\right]{[}^{3}O_{2}]}{{[}^{3}O_{2}]+\beta }=0,$$
where ξ, σ, δ, β, and g are specific PDT photochemical parameters. Comprehensive details are outlined in Table 1. The total [ROS]_rx_ produced during PDT can be calculated by integrating over the time course of the treatment using the measured ϕ, [S_0_] and [^3^O_2_]. For the sake of comparison, the total [ROS]_rx,calc_ was executed for all individuals by solving Equations (2)–(4) using measured ϕ and [S_0_], while the dynamic variation of [^3^O_2_](t) is determined from Equation (3) using an initial ground state oxygen concentration of ([^3^O_2_]_0_) based on measurement prior to PDT.

### 2.6. Photodynamic Therapy Efficiency Assessment

Kaplan–Meier curves were constructed to delineate the Local Control Rate (LCR) across all experimental groups [45]. After PDT, daily monitoring of tumor size was performed until attainment of a tumor volume equal to or surpassing 400 mm^3^. Within this context, the successful absence of tumor regrowth within a span of 90 days post-PDT was regarded as a “cure”. Tumor response data are presented in Table 2.

## 3. Results

Table 2 shows all treatment groups, the number of mice in each group, the treatment conditions, and the long-term treatment outcomes. The photochemical parameters for calculating [ROS]_rx_ are shown in Table 1. The fluence rate on the tumor surface was measured for each mouse at the beginning of PDT using an isotropic detector. Table 3 is a reproduction of Table 2 with those mice with [ROS]_rx_ > 1.1 mM excluded.

Figure 3 shows both the calculated and the measured oxygen concentrations ([^3^O_2_]) for all groups during the light illumination phase. Figure 4 shows the photosensitizer uptake at the beginning and end of PDT for the continuous and the two-part fractionation schemes. Figure 5 shows the comparison of evaluated total [ROS]_rx_ for each of the treatment groups including all mice (Figure 5a) and the mice with [ROS]_rx_ ≥ 1.1 mM excluded (Figure 5b). (Previous studies have shown that complete long-term cure can be achieved for single-fraction Photofrin-mediated PDT if [ROS]_rx_ reaches 1.1 mM [36]). Figure 6 shows Kaplan–Meier curves for all experimental conditions, including the control group. Calculated [ROS]_rx,calc_ and measured [ROS]_rx_ at 3 mm were compared as dosimetric quantities to assess their correlation with the long-term local control rate (LCR) for Photofrin-mediated PDT of RIF tumors across all fractionated groups in Figure 7a,b. The difference between the calculated and measured [ROS]_rx_ mainly arises from differences in oxygen concentration, as shown in Figure 3. In Figure 7c,d, the sigmoid fittings of LCR against the measured [ROS]_rx_ at 3 mm were plotted after the exclusion of mice with [ROS]_rx_ above 1.1 mM. We applied a sigmoid fitting using the equation y = 1/(1 + a × exp(−bx)), where a and b are the fitting parameters. The determination of the mean and standard deviations for ‘a’ and ‘b’ involved the proficient utilization of the global optimization toolbox available within Matlab_R2022b software (Curve Fitter). Subsequently, the grey area surrounding the fitting curve was obtained via Monte-Carlo (MC) statistical analysis using the uncertainties in parameters a and b to visualize the correlation between different dosimetric quantities and the long-term cure rate [13]. A narrower gray area indicates a better goodness of fit, which in turn suggests a stronger correlation between the LCR and the corresponding dosimetric parameter.

## 4. Discussion

Photodynamic Therapy (PDT) is a dynamic process involving three components—light, photosensitizer, and oxygen. This dynamic relationship presents challenges in comprehending the PDT process, thereby necessitating the exploration of avenues for enhancing treatment efficacy. Furthermore, uncertainties in microenvironments during the treatment increases the risks associated with PDT [46]. Currently, the widespread use of PDT is still limited due to the lack of long-term data, uncontrollable immune-related adverse effects, and the intricate nature of the procedure in the clinical application [47,48]. In this pursuit, our study adopts a streamlined two-segment fractionated treatment scheme, meticulously measuring explicit dosimetrical quantities ϕ, [*S*_0_], and [^3^O_2_] during the entire PDT treatment.

The measurements and results are summarized in Table 2 and Table 3. To account for the variation in light fluence and physiological conditions along the tumor depth, a tumor depth of 3 mm was employed. The values at 3 mm were determined through either direct measurement or simulation. For the MC simulation of the spatial distribution of light fluence rate ($\phi_{depth}$) in tumors, the mean tissue optical properties (μ_a_ = 0.9 cm^−1^, μ_s_′ = 8.4 cm^−1^) were utilized for all mice and $\phi_{3mm}$/$\phi_{air}$ = 0.83 ± 0.05 was obtained for a circular field of 1 cm diameter [49]. To validate the optical properties of tumor tissue in this study, the ratio of light fluence on the tumor surface ($\phi_{surface}$) to the in-air light fluence (ϕ_air_) at the same source-to-surface distance ($\phi_{air}$) was measured for selective mice, where the $\phi_{surface}$ includes the backscattering of the tissue. A mean ratio of $\phi_{surface}$/$\phi_{air}=$ 1.16 ± 0.04 was obtained. With the measured ratio and assuming a mean scattering coefficient (μ_s_′) of 8.4 cm^−1^, the absorption coefficient was calculated to be 0.79 ± 0.19 cm^−1^ using an expression for ϕ_surface_/ϕ_air_ = β(1 + 2R_d_), where β = 0.72 was provided by previous studies and the expression for $R_{d}=0.4843a^{\prime }\cdot (1+e^{-4.428\sqrt{1-a^{\prime }}})\cdot e^{-2.65\sqrt{1-a^{\prime }}}$, a′ = μ_s_/(μ_a_ + μ_s_′) is at the air-tissue interface [49]. The resulting ratio $\phi_{3mm}$/$\phi_{air}$ = 0.90 ± 0.10 [49] calculated for these optical properties (μ_a_ = 0.79 cm^−1^, μ_s_′ = 8.4 cm^−1^) is within 7% of the ratio using the mean tissue optical properties. This indicates that the effect of tissue optical properties on the light fluence rate at 3 mm in our mice is similar to the that of the previous study.

To assess the uncertainties of the in vivo fluorescence measurements, a comparison between ex vivo and in vivo measurements of the Photofrin concentration were made (Figure 2c). This was done within an alternate cohort of mice which were administered a range of Photofrin dosages. The symbols represent the average of three measurements, the error bars represent standard uncertainties of the mean. A linear fit depicted by a solid black line can be expressed as y = 0.979x (with R^2^ = 0.99). This result shows a favorable correlation between the in vivo and ex vivo datasets. This validates the integrity and accuracy of the in vivo photosensitizer measurements.

Figure 3 and Figure 4 show the in vivo measurements of [S_0_] and [^3^O_2_] undertaken during the PDT treatment. Notably, the initial ground state oxygen concentration ([^3^O_2_]_0_) exhibits a range spanning from 30 to 60 μM. This variability within [^3^O_2_]_0_ values is attributed to the inherent physiological conditions for each animal. The calculated mean [^3^O_2_]_0_ among all groups was 46.7 ± 21.9 μM, with a corresponding median value of 48.4 μM. Upon examination of Figure 3, an immediate drop in tissue oxygenation followed by a consistent elevation in tissue oxygenation at the beginning of the second fractionation is observed among all groups. This observed pattern is consistent with findings of previous investigations [50]. Our ROSED model-predicted oxygen consumption is shown as the gray dashed curve in Figure 3. The differences between the calculated and measured oxygen concentrations are attributed to the diverse biological and physiological differences among each animal. Notable factors include potential alterations in blood flow dynamics, a phenomenon underscored in the existing literature [51,52,53]. This shift in [^3^O_2_] dynamics during the latter stages of treatment serves to underscore the intricate and personalized nature of the treatment response. For the Photofrin concentration, even though each mouse was administered at the same dosage of 5 mg/kg, the initial Photofrin uptake (1 μM = 1.6 mg/kg) varied from 2.4 to 7.1 μM among mice prior to PDT. Theoretical prediction of uptake (lines) agrees with measurements (symbols). The mean photosensitizer concentration, [S_0_], was 4.8 ± 2.4 μM, with a corresponding median value of 4.9 μM. The temporal dependence of [S_0_] throughout all the PDT schemes is overlayed with theoretical predictions represented as solid lines. For the fractionation groups, reaccumulation of the photosensitizer becomes evident during the 2 h dark interval. Previous studies on tumor spheroids have examined Photofrin photobleaching and photoproduct accumulation, indicating a correlation with oxygen interaction [54,55]. The reason for this reaccumulation remains unknown.

Figure 5a shows that total [ROS]_rx_ was elevated for groups with different total treatment times (1800 s, 2400 s, 3000 s) but were approximately the same for the same total treatment time, within treatment uncertainties. Figure 5b is the same as Figure 5a except that the animals with [ROS]_rx_ ≥ 1.1 mM are excluded. Previous studies show complete long-term cure if [ROS]_rx_ reaches 1.1 mM even for single continuous PDT treatment [36]. Our current study indicates [ROS]_rx_ is not the reason for different outcomes observed among fractionated groups and between continuous and fractionated groups. As such, the mechanism underlying this fractionation-induced improvement remains elusive. Prior studies postulate either a vascular or immune response post-treatment due to fractionation [21,56].

The long-term treatment outcomes using the Kaplan–Meier curves are shown in Figure 6a. Table 2 shows details of treatment parameters and the 90-day local control rate (LCR). Fractionated treatments show consistently higher long-term survival (at 90 days) compared to their non-fractionated counterparts with the same total treatment time. For each subset, consisting of groups with the same total treatment times (1800, 2400, or 3000 s) or total light dose (135, 180, 225 J/cm^2^ using a nominal fluence rate of 75 mW/cm^2^), the treatment scheme characterized by the light fraction pairing of 800 + 1000 s, 1000 + 1200 s, and 1400 + 1600 s yielded the most favorable therapeutic outcomes among treatment groups with the same total treatment time. This corresponds to a long-term cure rate of 63%, 86%, and 100%. The first fraction is approximately 44%, 42%, and 47% of the total treatment time. This group of fractionated PDT is called “optimal fractionated scheme”. Note that the treatment scheme denoted as “1400 s + 1600 s” showcases a long-term cure rate of 100%. However, this achievement is accompanied by a pronounced level of observed toxicity (over 10% of lethality), thus rendering this combination impractical for clinical deployment.

Figure 6b demonstrates the long-term survival curves for Photofrin-mediated PDT for groups where the individuals with a total [ROS]_rx_ of over 1.1 mM were excluded. The previous study of single-fraction PDT shows that individuals with a total [ROS]_rx_ ≥ 1.1 mM have a 100% long-term LCR, a finding corroborated by our previous study [36]. Thus, to examine the effect of light fractionation alone, the individuals with a total [ROS]_rx_ higher than the threshold dose should be eliminated. The results after exclusion are shown in Figure 6b and Figure 7c,d, and Table 3. When we excluded individuals with [ROS]_rx_ levels surpassing the threshold dose, we observed that this had a minor impact on the overarching pattern but did lead to a reduction in the overall long-term LCR for all groups. The highest LCR drops to 57% and 80% for treatment using a total light fluence of 135 J/cm^2^ and 180 J/cm^2^. The LCR for group “1400 s + 1600 s” remains at 100% but the effective number of mice drops further to 1, which reduces the statistical significance. Remarkably, even after this exclusion, the most optimized treatment scheme remained “1000 s + 1400 s”, albeit with a reduced long-term LCR of 80%. This underscores the subtle interplay between treatment variables. The positive influence of light fractionation, when individuals with high [ROS]_rx_ were excluded, becomes notably evident.

In summary, we were able to increase the long-term survival without any toxicity from 20% (2400 s, total 180 J/cm^2^) under conventional single illumination to 80% in the optimized fractionated scheme (1000 + 1200 s, total 180 J/cm^2^) where we excluded animals with [ROS]_rx_ ≥ 1.1 mm. The increased survival is consistent with results obtained using bloodflow-based, fluence rate-modulated Photofrin-mediated PDT [33]. To study the correlation between dosimeter quantities and outcome, the correlations between dosimetric quantities ([ROS]_rx,cal_ and [ROS]_rx_) and the long-term treatment outcome, represented by local control rate (LCR), are shown in Figure 7. As is demonstrated by previous studies [13,57], a sigmoid curve (or threshold dose curve) can fit the data. Obviously, the correlation between calculated [ROS]_rx,cal_ and outcome is worse than that between measured [ROS]_rx_ and outcome, indicating that the calculated oxygen concentration is not enough to predict true tissue oxygen consumption during PDT. Figure 7c shows the relationship between measured [ROS]_rx_ and outcome when [ROS]_rx_ ≥ 1.1 mM are excluded, and this relationship has a slightly better R^2^ of 0.7221. The main reason for the large uncertainty is because of the known variation of outcome vs. the choice of the first and second treatment times. When only the optimal fractionation scheme is considered for each total treatment condition (Figure 7d), the uncertainty is now greatly reduced R^2^ = 0.9933.

To illustrate the improvement between 2-segment separated of 2 h dark period fractionated PDT treatment vs. continuous single-fraction PDT treatment with the same total treatment time, Figure 8 compares the [ROS]_rx_ response curve. The fitting for the optimal groups, represented by the blue dotted curve, clearly shifts to the left compared to the fitting curve of the single-fraction groups (black dashed curve). Note that in this figure, the result for single-fraction groups is shown for illustrative purposes. A comparison of the two fitting curves allows us to draw the conclusion that for mice with comparable levels of total [ROS]_rx_ production, those undergoing fractionated PDT tend to exhibit a higher long-term LCR, indicating an improved treatment outcome. Additionally, through this analysis, we have been able to ascertain the threshold [ROS]_rx_ for optimal fractionated Photofrin-mediated PDT, which stands at 0.78 mM. This value is notably lower than the threshold dose for single-fraction PDT, at 1.08 mM (the exact value may not be accurate).

## 5. Conclusions

This study centered on optimizing two-part (fractionated) PDT with a 2 h dark interval by varying the duration of light exposure to cohorts of mice RIF tumors. The primary objective was the 90-day tracking of tumor regrowth as a metric for long-term cure rate. PDT dosimetry includes real-time measurements of tissue oxygenation, photosensitizer uptake, and the light fluence rate to determine the reactive oxygen species concentration [ROS]_rx_. The study demonstrates that the introduction of a dark interval during treatment substantially increases treatment efficiency, even as other parameters remained relatively consistent. For treatments encompassing three total light fluences (135, 180, and 225 J/cm^2^), the optimal fractionation scheme is found for two conditions (135 and 225 J/cm^2^) denoted as “800 s + 1000 s” and “1000 s + 1400 s”. These produced long-term cure rates of 63% and 86%, respectively, without any toxicity. When the total light fluence increased to 225 J/cm^2^, a 100% long-term cure rate was achieved albeit with accompanying toxicity of death due to treatment, underscoring the clinical impracticality of this regimen. We noticed that the marked improvement in long-term survival was not correlated with the cumulative total [ROS]_rx_. The mechanism of this improvement is unknown, warranting further studies into the underlying mechanisms.

## Figures and Tables

**Figure 1 cancers-15-05682-f001:**
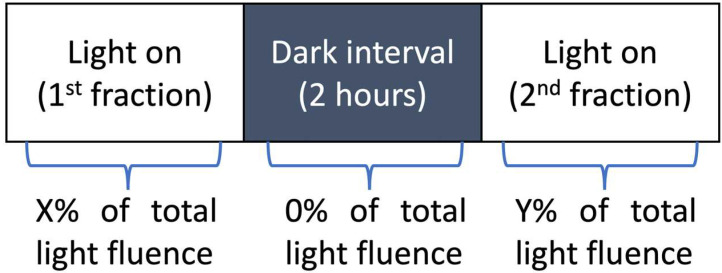
The general treatment plan design for fractionated treatment scheme. The values of X and Y are dependent on the specific treatment scheme. These values are controlled by varying the light fraction length, i.e., the illumination time of each light fraction.

**Figure 2 cancers-15-05682-f002:**
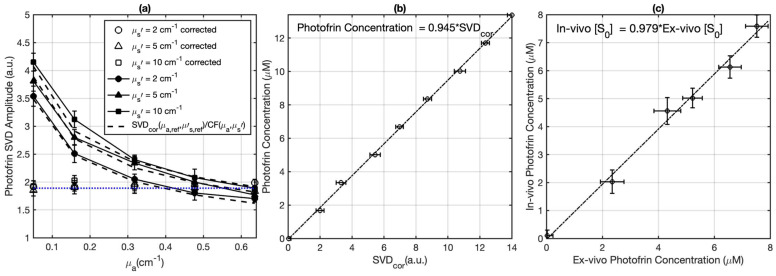
(**a**) Fluorescence singular decomposition (SVD) amplitude (arbitrary unit) for phantom experiments with different optical properties with the same Photofrin concentration (μ_a,ref_ = 0.64 cm^−1^, μ_s,ref_′ = 10 cm^−1^). The fitting result of the corrected SVD_cor_ is depicted by the blue dotted curve. The dashed curves illustrate the fitting obtained using SVD (μ_a,ref_, μ_s,ref_′)/CF (μ_a_, μ_s_′). (**b**) Photofrin concentration calibration curve. (**c**) The validation of in vivo measured photosensitizer concentration using the contact probe (based on fluorescence spectroscopy) against the ex vivo measured Photofrin concentration.

**Figure 3 cancers-15-05682-f003:**
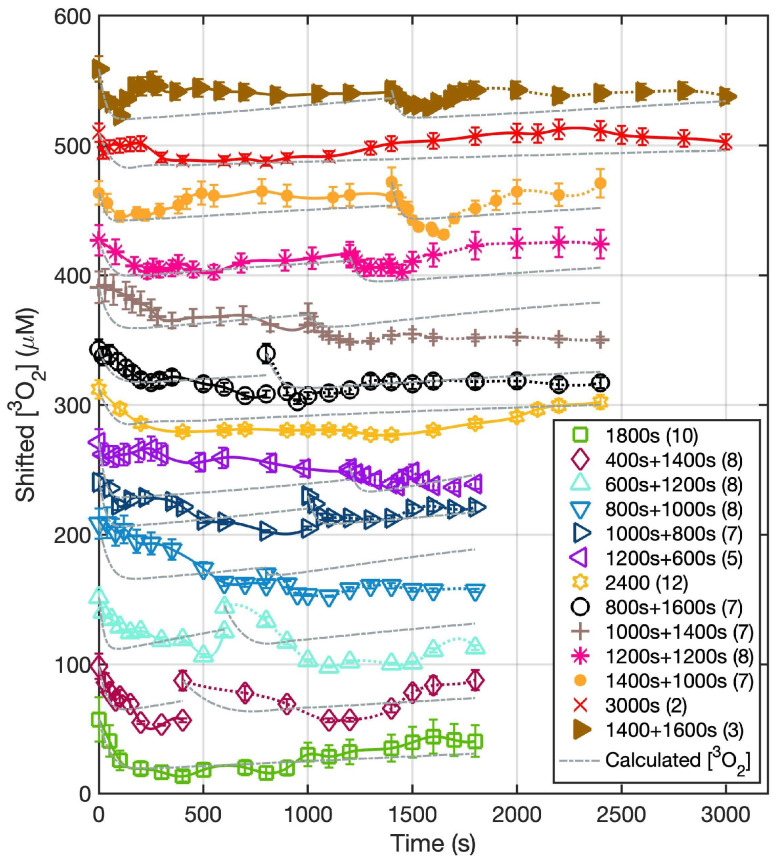
The temporal changes of ground state oxygen concentration versus time at a 3 mm depth for the treatment conditions. During the 2 h dark interval, no measurement was taken. The solid lines represent the measured temporal dependence of [^3^O_2_] over the course of treatment with the symbols representing [^3^O_2_] measured at discrete time points during PDT. The grey dash lines represent the calculated oxygen concentration ([^3^O_2_]_calc_) at a 3 mm depth calculated using Equations (2)–(4) and the parameters summarized in Table 1. The data for all groups, except for the “1800 s” group, are shifted by 100 on the y-axis for visualization.

**Figure 4 cancers-15-05682-f004:**
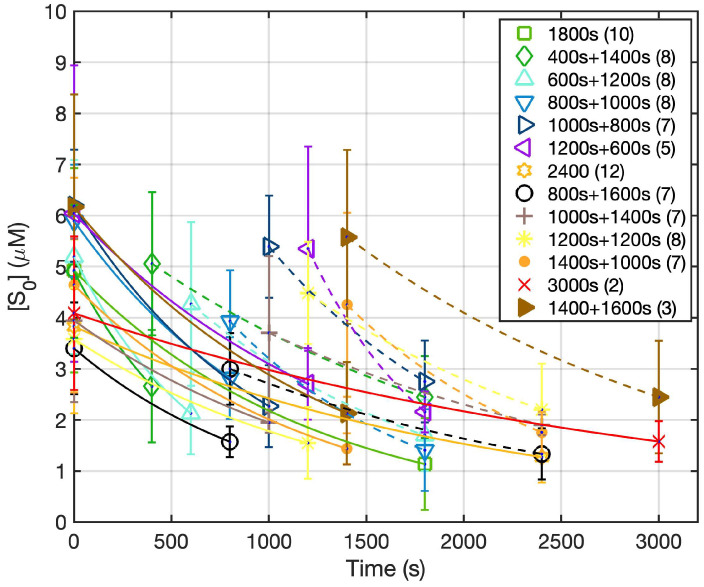
The dependence of Photofrin concentration vs. time for all treatment conditions. During the 2 h dark interval, no measurement was taken. The solid curves represent calculated Photofrin concentration based on Equations (2)–(4) during PDT treatment. The solid line represents the first fractionation and the dashed lines represent the second light fractionation. The symbols represent the measured PS concentration ([S_0_]).

**Figure 5 cancers-15-05682-f005:**
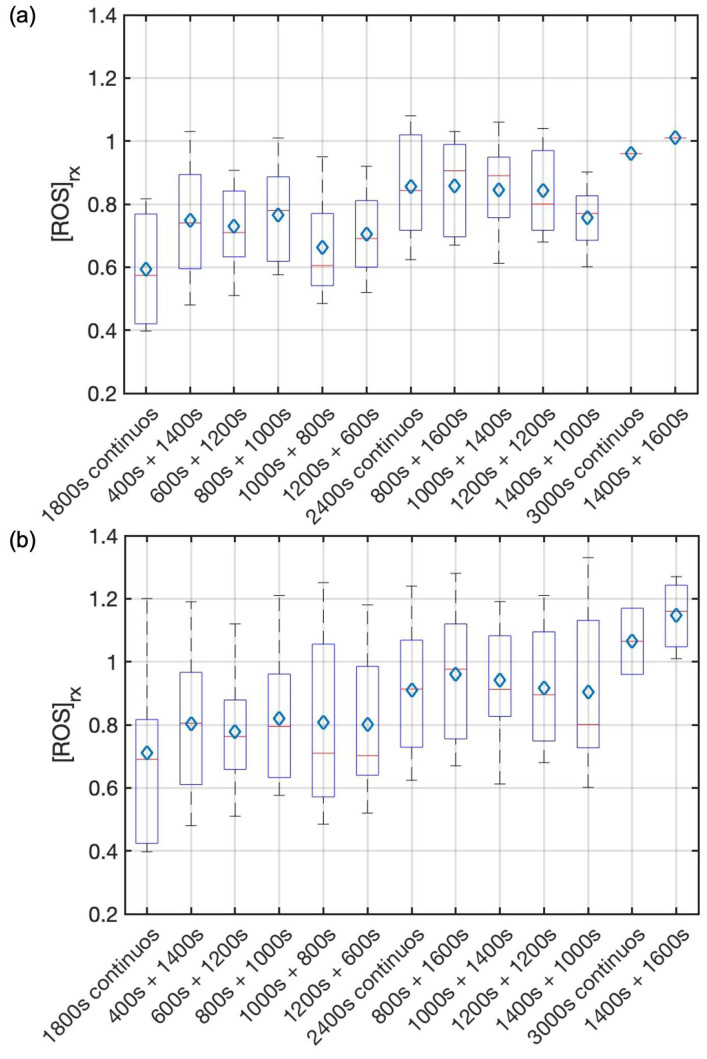
Comparison of total [ROS]_rx_ between different treatment schemes of (**a**) all mice and (**b**) mice with [ROS]_rx_ ≥ 1.1 mM excluded. On each box, the central mark and the central blue diamond indicate the median and the mean, respectively, and the bottom and top edges of the box indicate the 25th and 75th percentiles, respectively. The whiskers extend to the most extreme data points not considered outliers. The total amount of [ROS]_rx_ produced during PDT is similar for all fractionated groups. The improvement in treatment efficacy observed in the fractionated PDT groups is not correlated with the total reactive oxygen species ([ROS]_rx_) levels.

**Figure 6 cancers-15-05682-f006:**
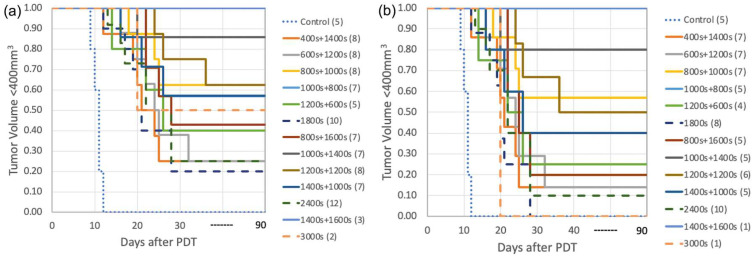
Survival curves for Photofrin-mediated PDT for groups with the total light fluences of (**a**) controls and 135 J/cm^2^, 180 J/cm^2^, and 225 J/cm^2^. The single-fraction groups as well as the controls are represented by dash and dotted lines. The observation period is 90 days for all groups. Mice were euthanized when tumor sizes reached 400 mm^3^ during the observation period. Among all groups, the best treatment scheme is “1000 s + 1400 s” with no obvious toxicity. (**b**) Long-term survival curves for Photofrin-mediated PDT for groups with total light fluences of 135 J/cm^2^ and 180 J/cm^2^. Mice with a total [ROS]_rx_ over 1.1 mM were excluded. Among all groups, the best treatment scheme remains “1000 s + 1400 s”.

**Figure 7 cancers-15-05682-f007:**
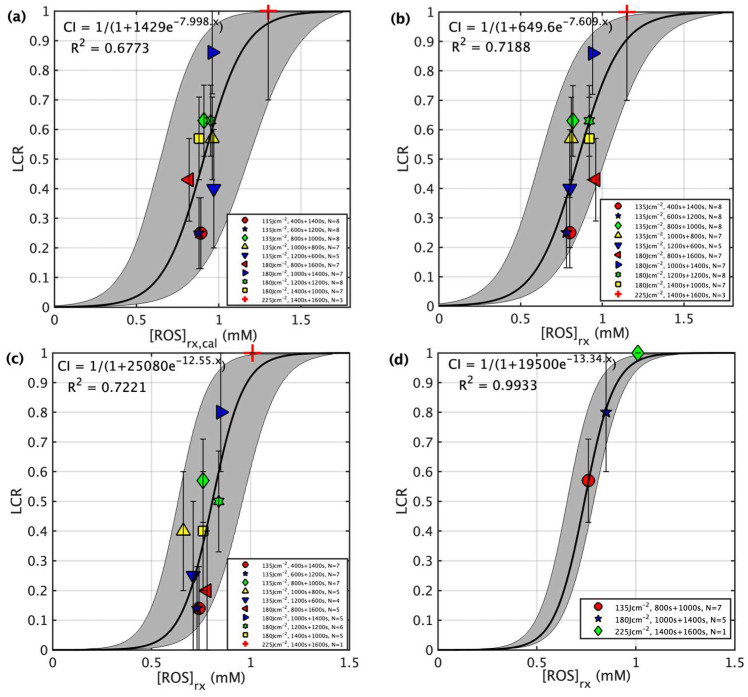
LCR plotted against the reactive oxygen species at a depth of 3 mm for four scenarios: (**a**) for calculated reacted reactive oxygen species concentration ([ROS]_rx,calc_) at a 3 mm depth calculated using Equations (2)–(4) and the parameters summarized in Table 1, (**b**) for measured reacted oxygen species ([ROS]_rx_) at a 3 mm depth for all individuals (Table 2), (**c**) for all fractionated groups (Table 3), and (**d**) for groups with optimized treatment schemes (Table 3). For (**c**,**d**), individuals with a total [ROS]_rx_ ≥ 1.1 mM are excluded. The solid lines show the best fit to the data with functional forms CI = 1/(1 + 1429e − 7.998x), 1/(1 + 649.6e − 7.609x), 1/(1 + 25,080e − 12.55x), and 1/(1 + 19,500 − 13.34x) with R2 = 0.6773, 0.7188, 0.7221, and 0.9933 for (a–d), respectively. The gray region indicates the upper and lower bounds of the fit with a 90% confidence level. As shown in (**c**), the resulting gray area for the optimized groups is significantly narrower, owing to the inclusion of the effects of light fractionation.

**Figure 8 cancers-15-05682-f008:**
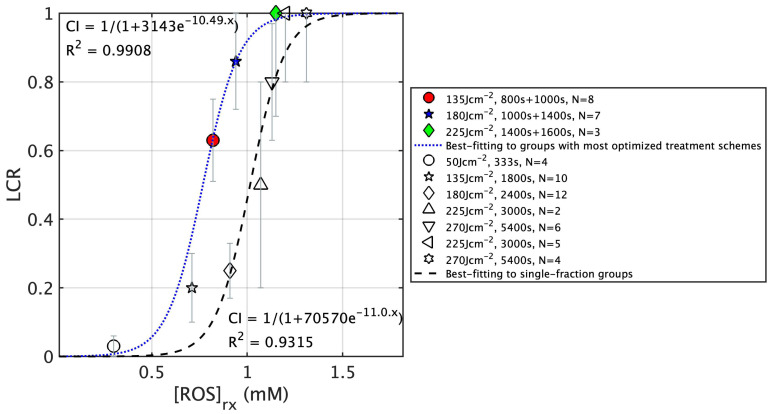
Sigmoid fittings plotted against measured reacted oxygen species ([ROS]_rx_) at a 3 mm depth for grouped data. The blue dotted curve and the dashed curve represent fitting for the groups with the most optimized fractionated treatment schemes and the single-fraction treatment schemes.

**Table 1 cancers-15-05682-t001:** Model parameters used in the macroscopic kinetics equations for Photofrin, obtained from the literature [39,40,41,42,43,44].

Parameter	Definition	Value
ξ (cm^2^s^−1^mW^−1^)	Specific oxygen consumption rate	3.7 × 10^−3^
σ (μM^−1^)	Specific photobleaching ratio	7.6 × 10^−5^
δ (μM)	Low-concentration correction	33
β (μM)	Oxygen quenching threshold concentration	11.9
g (μM/s)	Macroscopic oxygen maximum perfusion rate	0.76

**Table 2 cancers-15-05682-t002:** Summary of the treatment schemes, light fluence rate (ϕ), in vivo photosensitizer concentration ([S_0_]), tumor oxygen concentration ([^3^O_2_]), and treatment efficacies for each PDT group.

Groups	# of Mice	Treatment Schemes	In-Air Fluence Rate ^‡^ (mW/cm^2^)	[^3^O_2_]_0_ ^#^ (μM)	[S_0_]_0_ ^#^ (μM)	[S_0_]_end_ ^##^ (μM)	PDT Dose * (μMJ/cm^2^)	Mean [ROS]_rx,cal_ (mM)	Mean [ROS]_rx_ (mM)	LCR **
Control	5	0	0	-	0		0	0	0	0
1	8	400 s + 1400 s	74.4 ± 2.9	48.4 ± 9.1	4.9 ± 1.3	2.5 ± 1.0	401.6 ± 103.2	0.89 ± 0.19	0.80 ± 0.24	0.25 ± 0.12
2	8	600 s + 1200 s	73.5 ± 3.2	30.8 ± 5.6	5.2 ± 1.9	1.7 ± 0.7	329.9 ± 83.2	0.68 ± 0.15	0.78 ± 0.19	0.25 ± 0.12
3	8	800 s + 1000 s	75.5 ± 2.8	58.4 ± 12.4	4.9 ± 1.1	1.4 ± 0.8	328.2 ± 117.3	0.81 ± 0.18	0.82 ± 0.22	0.63 ± 0.12
4	7	1000 s + 800 s	76.0 ± 3.0	47.3 ± 7.5	6.2 ± 2.3	2.8 ± 1.1	467.1 ± 73.5	0.96 ± 0.26	0.81 ± 0.29	0.57 ± 0.14
5	5	1200 s + 600 s	75.8 ± 1.7	51.0 ± 10.4	7.1 ± 2.9	2.2 ± 0.4	475.1 ± 92.6	0.97 ± 0.25	0.80 ± 0.26	0.40 ± 0.20
6	10	1800 s	74.2 ± 2.6	57.4 ± 15.2	4.9 ± 2.0	1.1 ± 0.3	317.5 ± 56.2	0.79 ± 0.24	0.71 ± 0.30	0.20 ± 0.10
7	7	800 s + 1600 s	77.1 ± 2.8	52.7 ± 9.8	2.4 ± 0.9	1.3 ± 0.5	315.6 ± 93.4	0.82 ± 0.18	0.96 ± 0.22	0.43 ± 0.14
8	7	1000 s + 1400 s	72.6 ± 3.1	50.6 ± 12.4	4.0 ± 1.6	0.9 ± 0.2	339.0 ± 101.3	0.95 ± 0.14	0.94 ± 0.21	0.86 ± 0.14
9	8	1200 s + 1200 s	73.8 ± 3.3	46.8 ± 10.2	3.6 ± 1.0	2.2 ± 0.9	403.1 ± 73.1	0.86 ± 0.17	0.92 ± 0.19	0.63 ± 0.12
10	7	1400 s + 1000 s	74.1 ± 1.5	33.4 ± 8.9	4.6 ± 2.1	1.8 ± 0.4	409.4 ± 83.6	0.88 ± 0.21	0.92 ± 0.27	0.57 ± 0.14
11	12	2400 s	75.3 ± 2.0	42.0 ± 7.3	3.8 ± 1.7	1.3 ± 0.5	348.7 ± 38.5	0.83 ± 0.11	0.91 ± 0.20	0.25 ± 0.08
12	3	1400 s + 1600 s	74.6 ± 1.1	49.0 ± 5.3	6.2 ± 2.2	2.5 ± 1.1	705.9 ± 228.3	1.50 ± 0.10	1.15 ± 0.13	1.0 ± 0.3 ***
13	2	3000 s	75.3 ± 0.9	39.9 ± 3.6	4.1 ± 1.5	1.6 ± 0.4	508.0 ± 198.8	1.15 ± 0.13	1.07 ± 0.15	0.50 ± 0.50

^‡^ The measured mean light fluence rate in-air at the same height of the tumor surface. ^#^ The measured Photofrin and ground state oxygen concentrations prior to PDT. ^##^ The measured Photofrin after PDT is completed. * Calculation based on light fluence at a 3 mm depth, determined using the measured in-air light fluence rate, the mean tissue optical properties: μ_a_ = 0.9 cm^−1^ and μ_s_′ = 8.4 cm^−1^, and Monte Carlo (MC) simulation [14]. ** Local control rate at 90 days accessed based on V ≤ 100 mm^3^. *** High toxicity observed. Discontinued even though high LCR.

**Table 3 cancers-15-05682-t003:** Summary of the treatment schemes, light fluence rate (ϕ), in vivo photosensitizer concentration ([S_0_]), in vivo tumor oxygen concentration ([^3^O_2_]), and local control rate (LCR) for each PDT group with [ROS]_rx_ > 1.1 mM excluded.

Groups	# of Mice	Time (s)	[S_0_]_0_ (μM)	[S_0_]_end_ (μM)	PDT Dose (μMJ/cm^2^)	Mean [ROS]_rx_ (mM)	LCR
Control	5	0	0	0	0	0	0
1	7	400 s + 1400 s	4.3 ± 1.1	1.9 ± 1.0	318.4 ± 111.3	0.74 ± 0.20	0.14 ± 0.14
2	7	600 s + 1200 s	5.0 ± 1.6	1.7 ± 0.8	320.5 ± 72.4	0.73 ± 0.14	0.14 ± 0.14
3	7	800 s + 1000 s	4.3 ± 1.1	1.8 ± 0.9	297.5 ± 113.9	0.76 ± 0.16	0.57 ± 0.14
4	5	1000 s + 800 s	5.9 ± 2.0	2.4 ± 1.0	459.4 ± 82.5	0.66 ± 0.18	0.40 ± 0.20
5	4	1200 s + 600 s	6.3 ± 1.9	1.9 ± 0.6	394.4 ± 79.9	0.71 ± 0.17	0.25 ± 0.25
6	8	1800 s	4.5 ± 2.1	1.4 ± 0.6	286.5 ± 62.6	0.59 ± 0.18	0.00 ± 0.12
7	5	800 s + 1600 s	2.2 ± 0.7	0.9 ± 0.5	222.8 ± 101.0	0.78 ± 0.16	0.20 ± 0.20
8	5	1000 s + 1400 s	3.7 ± 1.5	1.1 ± 0.4	308.0 ± 103.1	0.85 ± 0.16	0.80 ± 0.20
9	6	1200 s + 1200 s	3.5 ± 0.9	2.0 ± 1.0	381.7 ± 68.4	0.84 ± 0.15	0.50 ± 0.17
10	5	1400 s + 1000 s	3.9 ± 2.4	1.7 ± 0.5	380.2 ± 89.2	0.76 ± 0.11	0.40 ± 0.20
11	10	2400 s	3.8 ± 1.9	1.2 ± 0.7	340.1 ± 47.1	0.86 ± 0.17	0.10 ± 0.10
12	1	1400 s + 1600 s	4.1	2.4	606.5 ± 210.9	1.01	1.0
13	1	3000 s	3.7	1.2	427.4 ± 202.7	0.96	0

## Data Availability

The data presented in this study are available on request from the corresponding author.

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
