# Peer review of "Fractionated Photofrin-Mediated Photodynamic Therapy Significantly Improves Long-Term Survival"

_cancers, 2023, doi:10.3390/cancers15235682_

Round 1

Reviewer 1 Report

Comments and Suggestions for Authors

      In this research, the authors evaluated the fractionated photofrin-mediated photodynamic therapy significantly improves long-term survival. Generally, it’s meaningful and interesting research. In my opinion, the current version of this manuscript fits the scope of the Cancers and could be accepted after minor revision.

My specific comments are in detail listed below:

1.     In this manuscript, the immune status after PDT or the capacity of PDT in killing tumor cells in vivo could be evaluated if possible.

2.     In the introduction part, the merits of PDT used in clinical and preclinical should be emphasized. Some references could be added to this part including 10.1016/j.jconrel.2022.11.004.

3.     All the figures are of low quality, which is hard to read. A better version should be added if possible. Besides, the references also should be checked.

4.     In the discussion part, the defects of PDT should be added. Besides, the author should predict or discuss the transformation barrier of PDT in clinical. Some references could be added to this part including 10.1002/adma.202206121.

5.     Some minor mistakes exist in this paper. The authors should polish it if possible.

6.     If possible, the authors may evaluate the capacity of PDT in killing tumor cells in vitro.  

Author Response

In this research, the authors evaluated the fractionated photofrin-mediated photodynamic therapy significantly improves long-term survival. Generally, it’s meaningful and interesting research. In my opinion, the current version of this manuscript fits the scope of the Cancers and could be accepted after minor revision.

My specific comments are in detail listed below:

  1. In this manuscript, the immune status after PDT or the capacity of PDT in killing tumor cells in vivo could be evaluated if possible.

We appreciate constructive suggestion by the referee but feel that this is currently beyond our ability to conduct the immune status study. We have emphasized that mechanism for the improved long-term (90 day) survival is still unknown (lines 56-57): “Nevertheless, the precise mechanism behind this improvement remains unknown.”

  1. In the introduction part, the merits of PDT used in clinical and preclinical should be emphasized. Some references could be added to this part including 10.1016/j.jconrel.2022.11.004.

      More description of the merit of PDT has been added into the introduction (lines 38-42). The reference suggested was inserted as Ref 7.

  1. All the figures are of low quality, which is hard to read. A better version should be added if possible. Besides, the references also should be checked.

Figures have been replotted with higher resolution. References were checked and modified.

  1. In the discussion part, the defects of PDT should be added. Besides, the author should predict or discuss the transformation barrier of PDT in clinical. Some references could be added to this part including 10.1002/adma.202206121.

      Revised in the text (line 373-378). More references were inserted (Ref 46-48), including the suggested one.

  1. Some minor mistakes exist in this paper. The authors should polish it if possible.

      We have checked the consistency and errors throughout the text in the revision.

Reviewer 2 Report

Comments and Suggestions for Authors

In this contribution the authors investigated the effect of fractionated PDT on mice using Photofrin as photosensitizer. They used a dosimetry model, ROSED for gauging the ROS produced during the treatment. From their experiments, the authors manage to demonstrate the improvement of using fractionated irradiation and established a threshold ROS concentration to reach these optimized results. This outcome may be used to enhance future PDT experiments.  I appreciated the very clear explanations provided by the authors about their methodological choices in the materials and methods section.

Author Response

We thank the referee for a positive evaluation.

Reviewer 3 Report

Comments and Suggestions for Authors

The authors provided a concise and clear abstract. The title of the article is well chosen Fractionated Photofrin-mediated Photodynamic Therapy Significantly improves long-term survival. The introduction clearly presents an outline of the topic and the current view of the problem. The results are impressive and clearly discussed. the aim of the study was a 90-day observation of tumor regrowth and PDT dosimetry includes real-time measurements of tissue oxygenation, photosensitizer uptake, and the light fluence rate to determine the reactive oxygen species concentration. The authors drew correct conclusions that provide opportunities for further research. please consider carefully editing the graphic of Fig.2

Sincerely

Author Response

Fig.2a has been revised to include the fitting results SVD(μa,ref, μs,ref’)/CF(μa,, μs’) (lines 249-251).  All figures (including Fig. 2) have been replotted for clarity.